# Multi-modal machine learning approach for early detection of neurodegenerative diseases leveraging brain MRI and wearable sensor data

Andrew Li[1], Jie Lian[2], Varut Vardhanabhuti [2,3]*

1 Department of Radiology, Queen Mary Hospital, Hong Kong SAR, China, 2 Department of Diagnostic Radiology, The University of Hong Kong, Hong Kong SAR, China, 3 Snowhill Science Limited, Hong Kong, Hong Kong SAR, China

* varv@hku.hk

## Abstract

Neurodegenerative diseases, such as Alzheimer's and Parkinson's Disease, pose a significant healthcare burden to the aging population. Structural MRI brain parameters and accelerometry data from wearable devices have been proven to be useful predictors for these diseases but have been separately examined in the prior literature. This study aims to determine whether a combination of accelerometry data and MRI brain parameters may improve the detection and prognostication of Alzheimer's and Parkinson's disease, compared with MRI brain parameters alone. A cohort of 19,793 participants free of neurodegenerative disease at the time of imaging and accelerometry data capture from the UK Biobank with longitudinal follow-up was derived to test this hypothesis. Relevant structural MRI brain parameters, accelerometry data collected from wearable devices, standard polygenic risk scores and lifestyle information were obtained. Subsequent development of neurodegenerative diseases among participants was recorded (mean follow-up time of 5.9 years), with positive cases defined as those diagnosed at least one year after imaging. A machine learning algorithm (XGBoost) was employed to create prediction models for the development of neurodegenerative disease. A prediction model consisting of all factors, including structural MRI brain parameters, accelerometry data, PRS, and lifestyle information, achieved the highest AUC value (0.819) out of all tested models. A model that excluded MRI brain parameters achieved the lowest AUC value (0.688). Feature importance analyses revealed 18 out of 20 most important features were structural MRI brain parameters, while 2 were derived from accelerometry data. Our study demonstrates the potential utility of combining structural MRI brain parameters with accelerometry data from wearable devices to predict the incidence of neurodegenerative diseases. Future prospective studies across different populations should be conducted to confirm these study results and look for differences in predictive ability for various types of neurodegenerative diseases.

## Author summary

Neurodegenerative diseases, like Alzheimer's and Parkinson's Disease, are a major health concern for the elderly worldwide. Our study aimed to improve the incidence prediction

**Data availability statement:** The data used for this work is from the UK Biobank Study, which can be accessed at https://www.ukbiobank.ac.uk/.

**Funding:** The author(s) received no specific funding for this work.

**Competing interests:** The authors have declared that no competing interests exist.

of such diseases by combining two types of data: MRI brain scans and accelerometry data collected from wearable devices. We analysed information from nearly 20,000 participants in the UK Biobank, including their MRI brain scans, accelerometry data, genetic risk scores, and lifestyle information. We recorded the development of neurodegenerative diseases among the participants and used machine learning algorithms to create prediction models. Our findings showed that the model incorporating all factors, including MRI brain parameters and accelerometry data, had the highest accuracy in predicting disease incidence. The results indicate that combining MRI brain scans with accelerometry data could be a powerful approach to predict the onset of neurodegenerative diseases. Further studies are needed to confirm these findings and explore their applicability to different types of neurodegenerative diseases.

## Introduction

Neurodegenerative diseases, which encompass conditions such as Alzheimer's Disease (AD), Parkinson's Disease (PD), and various types of dementia, impose a substantial burden on patients and healthcare systems worldwide [1–4]. In 2019, neurological diseases as a whole surpassed cardiovascular diseases as the largest cause of global burden [1]. Neurodegenerative diseases, more specifically, was found to be the second leading cause of death among neurological disorders (after stroke), and the third leading cause of disability-adjusted life years (DALYs) (after stroke and migraine) among the adult population [1,4,5]. The incidence of neurodegenerative disorders in 2019 was 7,236.38 in thousands, while the prevalence was found to be 51,624.19 in thousands, both showing over 240% increase compared with data from 1990 [1,4,5]. With the continued growth of an aging population, the burden of neurodegenerative diseases is expected to at least further double in the next two decades [4], and at least triple in 2050 to over 150 million cases worldwide [4]. The escalating incidence and profound impact of these diseases highlight the critical need for early detection to identify individuals in the prodromal stage, where interventions hold the greatest potential for efficacy.

### Structural MRI brain parameters as imaging biomarkers for neurogenerative diseases

Neuroimaging techniques have been heavily utilized, particularly in the last two decades, to screen for and monitor the progression of neurodegenerative diseases. Specifically, MRI has been one of the more popular imaging modalities employed by clinicians and researchers alike. Various MRI techniques, including structural MRI, functional MRI, diffusor tensor imaging and even neuromelanin-sensitive MRI, have been studied for potential use in diagnosing and prognosticating neurodegenerative changes [6]; however, structural MRI has remained the mainstay due to its relative ease of acquisition and strong evidence from the literature supporting its utility. Its strength lies in the ability to detect and differentiate atrophic morphological patterns in various neurodegenerative diseases. Scans of AD patients, for example, show characteristic disproportionate atrophy in the temporal lobe and medial parietal cortex [7]. Atrophic changes detected in structures of the medial temporal lobe, including the hippocampi, amygdala, cingulate cortex, parahippocampal gyrus and entorhinal cortex, have been consistently observed to be correlated with progression of AD [8–11].

Notably, hippocampal volume has been established as a reliable biomarker for AD diagnosis and severity, given ample evidence supporting its correlation with pathological findings and cognitive performance [12,13]. MRI volumetric measurements of the hippocampus were found to

be correlated with neurofibrillary tangles accumulation in the same brain region [14]. In addition, several studies demonstrated that the degree of hippocampal atrophy was associated with poorer performance on memory tasks [15,16]. Patients with frontotemporal dementia (FTD), on the other hand, often demonstrate disproportionate frontal, insular and anterior temporal lobe atrophy [17]. Interestingly, previous research has found atrophic changes to be less uniform in PD patients, only observing volume loss in parahippocampal gyrus, temporal gyrus and occipital lobe in severe cases [18,19]. However, more recent large-cohort research using deformation-based volumetry has provided evidence for a widespread multi-region atrophic pattern in Parkinson's patients that is readily detectable at the early stages of the disease [20]. Subcortical regions involved include all components of the basal ganglia, pedunculopontine nucleus, basal forebrain, hypothalamus, amygdala, hippocampus, parahippocampal gyrus, and the ventrolateral nucleus and pulvinar of thalamus. Cortical regions involved include the insula, occipital cortex Brodmann area 19, superior temporal gyrus, anterior cingulate cortex, premotor and supplementary areas, and lateral prefrontal cortex.

Besides volumetric measurements, cortical thickness measured on structural MRI has also been found to be predictive of the incidence and severity of neurodegenerative disorders. Cortical thickness at the entorhinal cortex, for instance, has been established as a sensitive biomarker for mild cognitive impairment and AD [21]. Moderate to severe PD cases have also been shown to have diffuse cortical thickness loss, especially those with disease duration of 5 years or more [18]. Interestingly, cortical thickness has also been demonstrated to reliably differentiate AD from FTD, in which AD patients consistently showed a greater degree of cortical thinning [22].

An additional imaging parameter detected on structural MRI that has strong associations with neurodegenerative diseases is white matter hyperintensities (WMHs). WMHs are one of the most common imaging features demonstrated on T2/FLAIR MRI sequences and are most commonly associated with cerebral small vessel disease. The most widely accepted pathological explanations for these lesions are chronic hypoperfusion and alterations in the blood-brain barrier [23], though other theories have been postulated, including demyelination, microglial activation, neuronal degeneration and even inflammation [24,25]. Numerous studies have shown an association between WMHs and a variety of neurodegenerative diseases, including MCI, AD, PD, Dementia with Lewy Bodies and FTD [26–32]. In fact, previous studies have shown that WMH burden is correlated with the severity of cognitive deficit in patients with neurodegenerative disease [33]. More recent studies have also lend support for WMH burden being associated with the severity of neuropsychiatric symptoms in this group of patients [34].

Though there is much evidence to support the utility of structural MRI brain parameters as biomarkers for neurodegenerative diseases, there are inherent imaging modality and patient-related factors that can affect the reliability of such brain parameters. Qualitative and quantitative assessment of MRI brain parameters, especially atrophy patterns and cortical thickness, may be affected by differences in scanner specifications and parameters, which can result in failure to detect disease progression or early changes. In addition, detection of these parameters may in some cases be only feasible with serial imaging, not just from imaging at one specific time point. Most importantly, patients with neurodegenerative diseases often have co-morbidities or multiple types of neurodegenerative diseases, causing difficulty in imaging interpretation and thus diagnosis [35,7]. White matter hyperintensities, for instance, are also present in patients with ischemic stroke. As a result, not all imaging biomarkers may be useful predictors for neurodegenerative disease incidence as a whole.

## Accelerometry data from wearable devices as a diagnostic and prognostic tool for neurodegenerative diseases

Neurodegenerative diseases are characterized by neuronal loss, which leads to not only cognitive deficits but also motor impairments. AD patients may present with a slower gait,

shorter stride length, and increased stride-to-stride variability [36,37]. PD, on the other hand, is typically associated with rigidity, tremors, and freezing, which can manifest in gait abnormalities such as lower walking speed, reduced step length, and impaired rhythmicity [37,38]. The advent of wearable devices containing accelerometers in the recent decade has made it increasingly accessible for researchers to track the motor and physical activity of patients across a 24-hour continuum. Initially, these wearable accelerometers were used to study physical activity levels, and large-scale studies found that self-reported as well as accelerometry-detected increase in total physical activity and less sedentary lifestyles are associated with a reduced risk of dementia and neurodegenerative diseases [39,40].

Research focus subsequently transitioned to more sophisticated data analysis, such as gait patterns, diurnal variation, and sleep, particularly in PD patients, as they often present with characteristic movement abnormalities, offering a window for early detection and tracking of disease severity [41]. Machine learning techniques have been employed to improve analysis and assess the predictive ability of accelerometry data. A small proof-of-concept study successfully employed support vector machine classifiers on wrist accelerometer data to detect PD with high accuracy (85%+/-15%), demonstrating the potential role of wearable devices in early detection [42]. Feature engineering methods, including epoch-based statistical feature engineering and the document-of-words method, were used to extract relevant predictive features from accelerometry data. A subsequent large-cohort study analysing UK Biobank data showed that wearable accelerometry features inputted into a Gaussian mixed model classifier could reliably diagnose PD, demonstrating an area under the curve (AUC) of 0.69 on gait data, 0.84 on low-movement data, and 0.85 on a fusion of both activities [43]. In addition, Schalkamp and colleagues (2023) demonstrated that accelerometry data from wearable devices, augmented by machine learning methods consisting of balanced random forests with Markov confusion matrices, were a significantly better predictor of established and prodromal PD than genetic markers, lifestyle factors, blood biochemistry or the presence of prodromal symptoms [44].

More recently, these machine learning techniques have also been used on accelerometry data collected from other neurodegenerative diseases. Accelerometry data has been employed to predict the diagnosis of both AD and PD. A large-cohort study conducted by Winer and colleagues (2024) also focused on UK Biobank data revealed that accelerometer-derived measures of activity levels, including interdaily stability, diurnal amplitude, and activity during the most active 10 hours, were predictive of both AD and PD incidence [45]. It is clear that while parameters derived from structural MRI may be reflective of cognitive deficits observed in patients with neurodegenerative diseases, patterns detected on accelerometry are more sensitive in capturing motor deficits associated with these diseases.

## Study objective

No previous study has examined the combined predictive ability of structural MRI brain parameters and accelerometry data on the incidence of neurodegenerative diseases. Given the unique insights that structural MRI and accelerometry data can provide on cognitive deficits and motor impairments, respectively, both modalities should be considered useful tools for predicting and monitoring neurodegenerative diseases.

Combining accelerometry data with established structural MRI brain parameters, therefore, may offer a multimodal approach to improve early detection of neurodegenerative diseases. The primary goal of this study is to investigate whether the combination of these two modalities does improve prediction of neurodegenerative disease incidence, compared with structural MRI brain parameters alone. To achieve this, we will leverage the rich phenotypic

and imaging data from the UK Biobank cohort to explore the predictive potential of this integrated approach.

# Results

## Demographics of study cohort

Table 1 provides an overview of the demographic characteristics of our study participants. Among the 19,793 individuals in our cohort, 56 were diagnosed with neurodegenerative diseases, while the remaining 19,737 were healthy. The mean age of the disease group was 69.9 years, significantly higher than that of the healthy group, which had a mean age of 64.1 years. Additionally, a higher proportion of men was observed in the disease group (60.7%) compared to the healthy group (45.4%).

Regarding lifestyle factors, a higher percentage of current smokers was found in the healthy group (63.4%) compared to the disease group, where a larger proportion were previous smokers (46.4%). A similar trend was observed in alcohol consumption, with more healthy participants reporting current drinking habits. Notably, individuals in the disease group reported spending more time on napping and daytime dozing (0.741 hours and 0.423 hours, respectively) compared to those in the healthy group (0.478 hours and 0.253 hours, respectively).

## Model performance

The comparison of model performances using the AUROC metric is listed in Table 2.

Model 1, which incorporated all modalities including MRI brain data, accelerometry data, PRS, and lifestyle information, achieved an AUC of 0.819. In comparison, Model 4, which excluded lifestyle information, attained an AUC of 0.759, indicating a decrease in predictive accuracy. Model 3, which omitted PRS information, yielded an AUC of 0.758, demonstrating a slight reduction in performance. Importantly, Model 2, excluding brain MRI data, exhibited the lowest AUC of 0.688.

The ablation studies demonstrated the varying contributions of each modality to the predictive performance. Notably, brain MRI data contributed the most to the final prediction,

**Table 1. Participants demographic data with lifestyle factors.**

|  | Healthy participants (n=19,737) | Disease participants (n=56) | P-values |
|---|---|---|---|
| Age (Years) | 64.1 (7.8) | 69.9 (6.2) | <0.0001 |
| Sex (Female percentage %) | 54.6% | 39.3% | 0.02 |
| Current Smoker | 63.4% | 51.8% | 0.113 |
| Previous Smoker | 33.6% | 46.4% |  |
| Current Drinking | 92.9% | 83.9% | 0.175 |
| Previous Drinking | 3.3% | 8.9% |  |
| Nap during day (hours) | 0.478 (0.604) | 0.741 (0.757) | 0.001 |
| Daytime dozing (hours) | 0.254 (0.485) | 0.423 (0.572) | 0.012 |

**Table 2. The comparison of model performances in terms of AUROC.**

| Model | AUROC |
|---|---|
| Model 1 (all modalities) | 0.819 |
| Model 2 (without brain MRI) | 0.688 |
| Model 3 (without PRS) | 0.758 |
| Model 4 (without lifestyle information) | 0.759 |

with its removal resulting in a significant decrease of 0.131 in AUC compared to Model 1. Conversely, the exclusion of PRS and lifestyle information led to relatively smaller reductions in AUC, decreasing by 0.06 and 0.061, respectively.

## Feature importance

Detailed analysis of the contribution of the various modalities provided further insight into the predictive power of each feature. We reported feature importance scores calculated using the XGBoost gain-based algorithm. Gain is defined as the improvement in accuracy brought by a feature in the model. Specifically, it measures how much a feature contributes to reducing the loss function when used for a split in the tree. A higher gain value indicates that the feature plays a more important role in improving the model's performance.

Fig 1 summarizes the top 20 important features identified by Model 1. Remarkably, 18 out of the 20 features were MRI-related, consisting of the grey matter volumes of various brain regions, as well as the total amount of white matter hyperintensities. In addition to these MRI features, two other features were based on accelerometry data: the average acceleration recorded by activity tracking during the time periods of 5 to 6 pm and 6 to 7 pm.

## Discussion

This study aimed to investigate the predictive ability of combining accelerometry data from wearable devices with MRI-based brain parameters in predicting the incidence of neurodegenerative diseases. Our results show that a statistical model (Model 1) that combined MRI brain features, accelerometry data from wearable devices, polygenic risk scores of neurodegenerative diseases, and lifestyle factors predicted more reliably the incidence of neurodegenerative

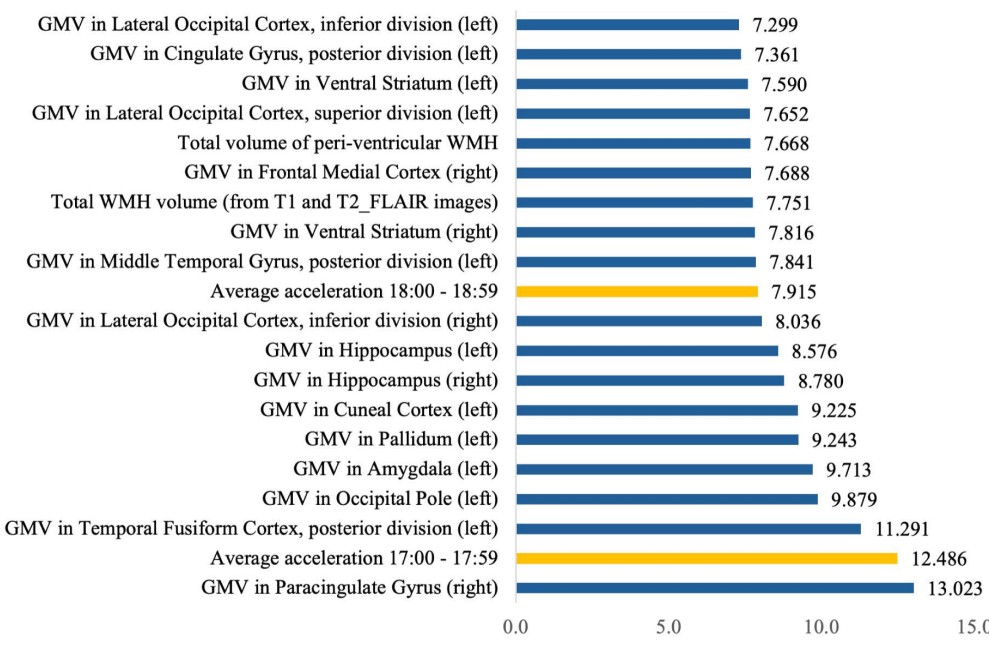

**Fig 1. Top 20 important features in predicting neurodegenerative disease as identified by Model 1 (Abbreviations: GMV = Grey Matter Volume; WMH = White Matter Hyperintensities).**

diseases in the UK Biobank cohort, compared with other statistical models (Model 2-4) that excluded either MRI brain data, PRS or lifestyle information. This finding not only reaffirms the importance of MRI brain parameters in predicting neurodegenerative diseases but also provides support for the combination of MRI brain parameters and accelerometry data from wearable devices in improving the early detection of the onset of neurodegenerative diseases.

To be specific, 18 out of the 20 most important predictive features for neurodegenerative diseases as derived from Model 1 were MRI brain parameters. They include total white matter and total periventricular white matter hyperintensities, as well as the grey matter volumes of bilateral hippocampi, left temporal fusiform cortex, bilateral ventral striatum, left posterior cingulate gyrus, left cuneal cortex, left pallidum, left amygdala, left middle temporal gyrus, right paracingulate gyrus, bilateral lateral occipital cortices/left occipital pole and right frontal medial cortex. It was anticipated that the majority of identified predictive features in the model were MRI brain parameters, given that MRI features have long been considered key diagnostic and prognostic biomarkers for neurodegenerative diseases. The improved predictive ability of the statistical model that incorporated both MRI brain parameters and accelerometry data suggests clear added value that accelerometry data brings to the detection of neurodegenerative diseases.

The identified brain regions that contributed significantly to the predictive ability of the model are consistent with previous literature on neurodegenerative diseases. Total periventricular white matter hyperintensities have been consistently linked with small vessel disease, which is found in most neurodegenerative diseases [27–29]. Multiple studies have reported significantly increased volume of white matter hyperintensities in patients with AD and PD [27–33].

As discussed, volume loss of bilateral hippocampi is well-established as a biomarker for mild cognitive impairment and AD [12–16]. An MRI volumetric study, in fact, demonstrated focal atrophy of the CA1 subfield in the early (predementia or even preclinical) stages of AD, before widespread atrophy of the whole hippocampus begins at the dementia stage [46]. In PD, the severity of hippocampal atrophy is correlated with the severity of cognitive impairment [47,48]. Volume loss in the temporal fusiform cortex, which has been implicated in the formation and storage of semantic information, is primarily associated with semantic dementia rather than AD or PD [49,50]. Volume reduction of the ventral striatum is a recognized biomarker for PD, as it is well established that loss of dopaminergic neurons in the striatal network forms the pathological basis of the disease [51]. Multiple studies have also reported significant atrophy in striatal structures in AD patients, potentially secondary to the degeneration of connected structures, such as the hippocampus [52,53].

A few studies have demonstrated atrophy of the posterior cingulate gyrus in AD and PD patients, though the underlying mechanism remains unclear [54–57]. Some have attributed the finding to the fact that the region is part of the default mode network, which is often impaired in the case of neurodegenerative diseases [55]. The cuneal cortex, known to be involved in cognitive and emotional coordination, has been shown to demonstrate early cortical atrophy in PD patients with cognitive impairment [58–60]. Volume loss of the pallidum, whose key subregions include the thalamus and globus pallidus, has also been associated with neurodegenerative disease progression. A study showed thalamic volume loss as one of the first signs of cognitive decline during early mild cognitive impairment, though no volume loss is observed during further progression to AD [61]. Other studies have also demonstrated thalamic volume loss in the early stage of PD [62,63], which could be attributed to the region's strong connectivity with the striatal region [63].

The amygdala, regarded as the center for emotional information processing, shows significant volume loss in PD patients, especially in those with depressive and anxiety symptoms

[64]. Some studies have also reported amygdala volume loss in AD patients [65,66]; atrophy of the amygdala could also be due to the regions' strong connectivity with the hippocampus. The middle temporal gyrus, an important site for memory processing [67–70], has been shown to be one of the first neocortical regions in the temporal lobe to show volume loss in AD patients [67,68]. The paracingulate gyrus, responsible for mentalizing, inhibitory control, and guiding motor actions, exhibits prominent atrophy in frontotemporal dementia patients [70,71].

The occipital cortex, specifically the lateral occipital cortex, known to be involved in object recognition and visuospatial processing, has also been shown to undergo a reduction in volume among PD patients [72]. The frontal medial cortex has also been widely implicated in the progression of various neurodegenerative diseases. Overall frontal lobe volume reduction has been shown in late-stage PD by multiple studies [57,73–78].

Our study is the first to demonstrate that accelerometry data from wearable devices can serve as a significant predictor of neurodegenerative disorders as a whole. Previous studies have primarily focused on the predictive ability of accelerometry data in tracking and predicting PD. Winer and colleagues (2024) demonstrated that 24-hour rhythm integrity measured by wearable devices was an associated risk of AD, PD and cognitive decline [45]; however, it did not explore its association with all neurodegenerative diseases. As discussed, a likely explanation for the predictive ability of accelerometry data on all neurodegenerative diseases is the universal effect of neurodegenerative diseases on gait, which could be indirectly reflected in accelerometry [79]. Our study hence highlights the broad potential applicability of wearable device data in not only tracking but also predicting a wide range of neurodegenerative diseases.

Despite the significant findings of this study, several limitations should be acknowledged. Firstly, this study is an association study and cannot establish a causal relationship between MRI brain parameters and accelerometry data with the incidence of neurodegenerative diseases. Secondly, the UK Biobank cohort used in this study contains only a small number of positive cases for neurodegenerative diseases, resulting in an imbalanced dataset that could introduce statistical bias in the analysis. Thirdly, the study was performed on data from the UK Biobank cohort with a mean follow-up period of 5.9 years. A longer period of follow-up is always desirable as well as multiple time points longitudinal tracking, albeit may not be feasible at large scale. Finally, future prospective studies will be helpful, particularly focusing on specific neurodegenerative diseases (e.g. Alzheimer's or Parkinson's disease, for example) to also reduce the impact of different disease heterogeneity. Testing in different populations would also ensure the reproducibility of these findings. Nevertheless, further research in this area holds promise for improving the early detection, diagnosis and management of these debilitating conditions.

In conclusion, this study provides evidence for the strength of combining MRI brain parameters and accelerometry data from wearable devices to predict the incidence of neurodegenerative diseases. Using a cohort of 19,793 participants from the UK Biobank, free of neurodegenerative disease at baseline and followed for an average of 5.9 years, we employed the XGBoost machine learning algorithm to develop prediction models. The comprehensive model, incorporating MRI parameters, accelerometry data, polygenic risk scores, and lifestyle information, achieved the highest AUC value of 0.819. In contrast, the model excluding MRI parameters had the lowest AUC value of 0.688. Feature importance analysis highlighted that 18 of the top 20 predictors were MRI brain parameters, while 2 were from accelerometry data, emphasising the critical role of MRI parameters in predicting neurodegenerative diseases. Future studies should explore the predictive ability of other MRI brain parameters, such as cortical thickness and functional connectivity, in conjunction with accelerometry data from wearable devices. Additionally, studying cohorts with more time points and longer follow-up

would allow for a better assessment of the temporal association between MRI brain parameters and accelerometry data with neurodegenerative disease incidence. It would also be valuable to investigate the differences in predictive ability for various types of neurodegenerative diseases, such as AD, PD and dementia.

## Methods

### Ethical approval

Ethical approval was obtained for this study from the UK Biobank study (UK North West Multi-Centre Research Ethics Committee under reference 11/NW/0382) and the University of Hong Kong (UW-20814). Written informed consent was secured from all individuals participating in the study. This research was conducted using the UK Biobank Resource Application Number 78730.

### Participants and study design

This study utilized data from the UK Biobank, a comprehensive dataset containing biological and medical data from over 502,000 participants [80]. Our initial cohort consisted of 48,457 participants, who had both T1 and T2 weighted brain MRI data available. From this initial pool, exclusions were made for individuals lacking accelerometry data (N=28,560), those without an official standard PRS provided by the UK Biobank (N=30), and those with missing lifestyle information including smoking habit, alcohol consumption, duration of daytime napping, and duration of daytime dozing (N=74). These exclusions yielded a final sample size of 19,793 participants, as illustrated in Fig 2.

To identify individuals with neurodegenerative diseases (AD, PD and dementia), we employed algorithms developed by the UK Biobank Outcome. Cases were defined as individuals diagnosed with any of these diseases at least one year after their image screening. The mean follow-up time for our cohort is 5.9 years. This process identified 56 positive cases, comprising 19 with PD, 13 with AD, and 40 with dementia. Notably, 10 participants were diagnosed with all three diseases, while 27 exhibited two of the specified diseases.

### Brain MRI pre-processing

The UK Biobank's MRI brain data were acquired from three imaging centers (Cheadle, Newcastle, and Reading) using 3T Siemens Skyra scanners.

The imaging time period for our participants was between 05/2014 and 03/2020. We used the processed brain images as our input data. Specifically, for T1-weighted brain MRI analysis, we utilized the 139 regional grey matter volumes (GMV) generated by FSL FAST.

For T2-weighted MRI analysis, we focused on 5 features related to white matter hyperintensities (WMH) volumes. T2-weighted MRI is a valuable tool for detecting WMH in neurodegenerative diseases. WMH appears as areas of increased signal intensity on T2-weighted and fluid-attenuated inversion recovery (FLAIR) MRI scans.

For reproducibility and comparability, all segment volumes' information was generated by the UK Biobank using field IDs 1101 and 112. Detailed information can be found in S1 and S2 Tables.

### Polygenic risk score data

Polygenic Risk Scores (PRS) are increasingly recognized for their potential in understanding and predicting neurodegenerative diseases. A PRS quantifies an individual's genetic predisposition to a disease based on the cumulative effect of numerous single nucleotide polymorphisms (SNPs) identified through genome-wide association studies (GWAS). In the context of neurodegenerative

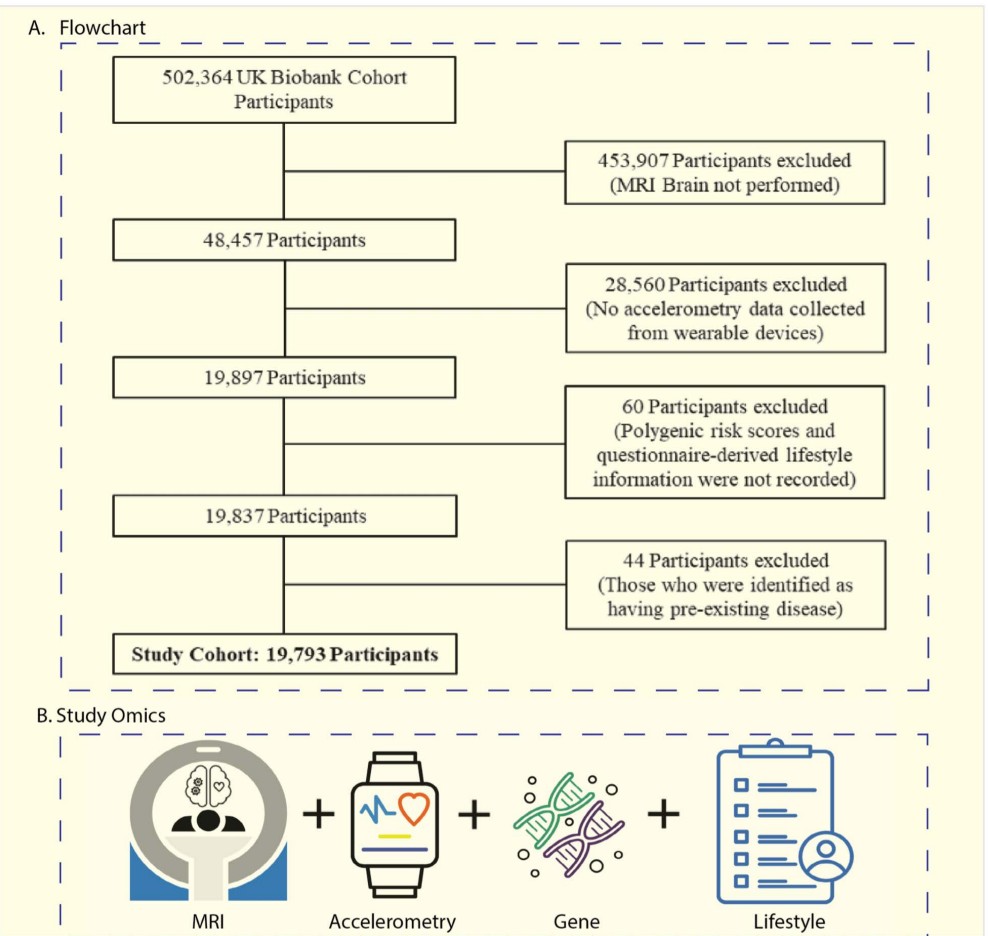

**Fig 2. (A) Flowchart of Patient Inclusion Process; (B) Study omics including MRI, accelerometry data, genetic data and lifestyle information.**

diseases, such as AD and PD, PRS can correlate with disease risk and progression, providing insights into genetic liability. For reproducibility and comparability, all PRS scores were calculated by the UK Biobank using field IDs 301. Detailed information can be found in S3 Table.

## Accelerometry data

To investigate the association between physical activity patterns and neurodegenerative disease incidence, we focused on participants' accelerometry measurements. Measurements were collected via wrist-worn accelerometers, with the primary data collection period spanning from June 2013 to January 2016. Derived accelerometry data provided key metrics, including the average proportion of time spent engaging in light activity, moderate-vigorous activity, sedentary behavior, and sleep per day. Also, average accelerations measured during the activity monitor's wearing period were used to further characterize participants' physical activity levels. Please refer to S4 and S5 Tables for details.

## Lifestyle data

Neurodegenerative disorders are greatly affected by factors related to lifestyle, such as smoking, alcohol consumption, and sleep patterns. Studies suggest that people who adhere to a

healthy lifestyle, which includes abstaining from smoking, consuming alcohol in moderation, engaging in regular physical activity, and getting sufficient sleep, have a reduced likelihood of developing diseases. We included smoking status, drinking status and frequencies, nap time during day, and daytime dozing/sleeping and lifestyle factors.

### Input modalities

This study employed 4 modalities: (1) MRI: We utilized T1-weighted brain MRI segments to assess 139 regional grey matter volumes (GMVs), and 5 features related to white matter hyperintensities (WMH) volumes generated from T2-weighted MRI to focus. (2) Polygenic Risk Scores (PRS): We used 39 common diseases' PRS scores. (3) Accelerometry Data: We utilized data reflecting the average proportion of time spent in various activities, including light activity, moderate-to-vigorous activity, sedentary behavior, and sleep per day, as well as average accelerations. (4) Lifestyle Information: Smoking status, drinking habits, and sleep patterns were included as input.

### Classification performance

To assess the predictive power of disease incidence, we devised an experimental setup consisting of four models:

- Model 1: Utilizing all modalities of interest, including MRI brain data, accelerometry data, PRS, and lifestyle information as input.

- Model 2: Exclusively using accelerometry data, PRS, and lifestyle information.

- Model 3: Utilizing MRI brain data, accelerometry data, and lifestyle information.

- Model 4: Utilizing MRI brain data, accelerometry data and PRS.

XGBoost, an advanced machine learning algorithm, has shown promising results in medical classification tasks. It outperforms traditional methods like logistic regression in predicting myocardial infarction, achieving higher ROC scores [81]. XGBoost demonstrates robustness in handling missing data, maintaining high F1-scores even with 90% missing values in breast cancer and heart failure datasets [82]. The algorithm excels at managing complex, high-dimensional data, as evidenced by its superior performance in cancer stage prediction using multi-omics data [83]. Importantly, XGBoost offers interpretability through feature importance ranking and SHAP values, bridging the gap between medicine and data science [83,84]. Its ability to handle data complexities and provide interpretable results makes XGBoost a valuable tool for medical practitioners in diagnosis and treatment decision-making [84].

To evaluate the model's performance, the dataset was randomly split into 80% for training and 20% for testing. The model was trained and tuned using the training dataset with the strategy of five-folder-cross-validation. We applied gride-search methods to find the best hyper-parameters for each model, including: "subsample", "reg_lambda", "reg_alpha", "n_estimators", "min_child_weight", "max_depth", "gamma", "eta" and "colsample_bytree".

We applied the Area Under the Receiver Operating Characteristic (AUROC) curve as our evaluation metric. AUROC measures a model's ability to distinguish between classes across all possible classification thresholds. Given the significant class imbalance in this dataset, using AUROC for unbalanced classification tasks in disease classification offers a robust, comprehensive, and interpretable assessment of model performance. The results can be found in S6 Table.

## Statistical analysis

Correlation analyses between the disease group and healthy controls were conducted using T-tests. All statistical analyses were performed using Python (version 3.10), with significance set at a two-sided level of 5%. Performance metrics were presented with 95% confidence intervals to accurately reflect the variability in the data and the precision of our estimates.

## Supporting information

**S1 Table. T1 Brain MRI features and the related field IDs in UK Biobank.**
(DOCX)

**S2 Table. T2 Brain MRI features and the related field IDs in UK Biobank.**
(DOCX)

**S3 Table. RPS scores and the related field IDs in UK Biobank.**
(DOCX)

**S4 Table. Accelerometry average data and the related field IDs in UK Biobank.**
(DOCX)

**S5 Table. Accelerometry-derived data and the related field IDs in UK Biobank.**
(DOCX)

**S6 Table. Comparison with Common Models.**
(DOCX)

## Author contributions

**Conceptualization:** Varut Vardhanabhuti.

**Data curation:** Andrew Li, Jie Lian, Varut Vardhanabhuti.

**Formal analysis:** Jie Lian, Varut Vardhanabhuti.

**Investigation:** Varut Vardhanabhuti.

**Methodology:** Andrew Li.

**Supervision:** Varut Vardhanabhuti.

**Visualization:** Andrew Li.

**Writing – original draft:** Andrew Li, Varut Vardhanabhuti.

**Writing – review & editing:** Andrew Li, Jie Lian, Varut Vardhanabhuti.

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
