## [Decision Letter · Decision Letter 0]

2 Aug 2024

PDIG-D-24-00170

Multi-modal Machine Learning Approach for Early Detection of Neurodegenerative Diseases Leveraging Brain MRI and Wearable Sensor Data

PLOS Digital Health

Dear Dr. Vardhanabhuti,

Thank you for submitting your manuscript to PLOS Digital Health. After careful consideration, we feel that it has merit but does not fully meet PLOS Digital Health's publication criteria as it currently stands. Therefore, we invite you to submit a revised version of the manuscript that addresses the points raised during the review process.

Please submit your revised manuscript within 60 days Oct 01 2024 11:59PM. If you will need more time than this to complete your revisions, please reply to this message or contact the journal office at digitalhealth@plos.org. Please include the following items when submitting your revised manuscript:

We look forward to receiving your revised manuscript.

Kind regards,

Md. Mehedi Hassan

Academic Editor

PLOS Digital Health

Additional Editor Comments (if provided):

Authors need to focus on paper presentations, ensuring the outcomes are clearly rewritten to highlight the proper findings from this study.

I recommend addressing all reviewers' comments thoroughly and resubmitting the paper to the journal.

Reviewers' comments:

Reviewer's Responses to Questions

**Comments to the Author**

1. Does this manuscript meet PLOS Digital Health’s publication criteria ? Is the manuscript technically sound, and do the data support the conclusions? The manuscript must describe methodologically and ethically rigorous research with conclusions that are appropriately drawn based on the data presented.

Reviewer #1: No

Reviewer #2: Yes

Reviewer #3: Partly

Reviewer #4: No

2. Has the statistical analysis been performed appropriately and rigorously?

Reviewer #1: No

Reviewer #2: No

Reviewer #3: No

Reviewer #4: No

3. Have the authors made all data underlying the findings in their manuscript fully available (please refer to the Data Availability Statement at the start of the manuscript PDF file)?

Reviewer #1: Yes

Reviewer #2: Yes

Reviewer #3: No

Reviewer #4: No

4. Is the manuscript presented in an intelligible fashion and written in standard English?

PLOS Digital Health does not copyedit accepted manuscripts, so the language in submitted articles must be clear, correct, and unambiguous. Any typographical or grammatical errors should be corrected at revision, so please note any specific errors here.

Reviewer #1: No

Reviewer #2: Yes

Reviewer #3: No

Reviewer #4: No

5. Review Comments to the Author

Please use the space provided to explain your answers to the questions above. You may also include additional comments for the author, including concerns about dual publication, research ethics, or publication ethics. (Please upload your review as an attachment if it exceeds 20,000 characters)

Reviewer #1: The paper titled "Multi-modal Machine Learning Approach for Early Detection of Neurodegenerative Diseases Leveraging Brain MRI and Wearable Sensor Data" represents a Multi-modal Machine Learning Approach. However, several areas require further development to enhance its scientific impact:

1. Improve the abstract by clearly outlining the specific aims, methods, key results, and conclusions. Mention specific neurodegenerative diseases being targeted.

2. Provide more detailed background on the burden of neurodegenerative diseases, including current statistics on incidence and prevalence.

3. Include a separate Literature Review section to include recent advances in MRI biomarkers and wearable sensor technology for neurodegenerative diseases.

4. Clearly articulate the research gap your study is addressing and how it advances current knowledge.

5. Explicitly state the hypothesis being tested in the study.

6. Specify which MRI brain parameters were selected and why. Include details on how these parameters are linked to neurodegenerative diseases.

7. In the Classification Performance section, explain the calculation of polygenic risk scores (PRS) and their relevance to neurodegenerative diseases.

8. Describe the lifestyle information collected and its potential impact on neurodegenerative disease progression.

9. Detail the data preprocessing steps for both MRI and accelerometry data to ensure reproducibility.

10. Justify the choice of XGBoost as the machine learning algorithm and explain any hyperparameter tuning performed. Implement at least five more machine learning algorithms and compare the performance.

11. Add a Performance Matrices section to discuss the evaluation metrics used (AUC, accuracy, etc.) and why they were chosen.

12. In the Feature Importance section provide a more detailed explanation of the feature importance analysis and its implications. Such as visualizing the performance with and without selecting important features.

13. In the Statistical Analysis section include visual graphs to present statistical analyses conducted to validate the findings, such as significance testing and confidence intervals.

14. Add a section Comparison with Existing Methods to compare your multi-modal model with other existing models in the literature to highlight its advantages.

15. Discuss the limitations of your study, including any potential biases or confounding factors.

16. Provide more concrete suggestions for future prospective studies, including specific types of neurodegenerative diseases to be investigated.

17. Include more figures and tables to increase reader understandability. Ensure all figures and tables are clear, and well-labeled.

18. The paper lacks a well-structured outline, making it difficult for readers to follow the flow of information. It needs clear, distinct sections for the Introduction, Literature Review, Methods, Dataset, Data Preprocessing, Performance Metrics, Result Analysis, Comparison with Existing Work, Discussion, and Conclusion. Each section should be clearly defined and logically organized to enhance readability and comprehension.

By addressing these comments, the paper can be significantly improved in terms of scientific precision, clarity, and overall impact on the field of neurodegenerative disease research.

Reviewer #2: The primary goal of this study is to investigate the combination of accelerometry data from wearable devices and MRI-based brain parameters in predicting the incidence of neurodegenerative diseases. However, the authors should concentrate on the following issues before resubmission:

(i) The authors should show the contributions in point-by-point basis.

(ii) The authors should add a literature review section as well.

(iii) The authors should elaborate the technical method applied here in the study.

(iv) The authors should also show the results of more analysis to come to the conclusion.

Reviewer #3: Suggestions for Improvement formatting

Consolidate Related Questions:

Group similar questions together to avoid redundancy.

Use Numbering or Bullet Points:

Organize questions with clear numbering or bullet points.

Clear Headers or Titles:

Provide headers for different sections of questions to enhance readability.

Structured Responses:

Respond to each question in a structured format, corresponding to the numbering or bullet points used for questions.

Seek Clarification if Needed:

Request clarification on unclear or repetitive questions to streamline communication.

Novelty

What is the primary focus of the study regarding neurodegenerative diseases?

How many participants were involved in the study cohort, and where were they sourced from?

What types of data were collected from the participants?

How was the incidence of neurodegenerative diseases defined and recorded during the study?

Which machine learning algorithm was used, and what was its purpose in the study?

What was the highest achieved AUC value, and which model configuration achieved it?

According to the feature importance analysis, what were the top contributors to predicting neurodegenerative disease incidence?

Major concern

Detailed Questions Raised:

Evidence of Improvement:

What specific empirical evidence or studies demonstrate a clear improvement in predictive accuracy when combining accelerometry data and MRI-based brain parameters compared to using either method alone?

How do these studies control for potential confounding variables and biases that may influence the observed outcomes?

Limitations of MRI Parameters:

Given the heterogeneous nature of neurodegenerative diseases, what are the acknowledged limitations of MRI-based parameters in accurately capturing disease progression and early pathological changes?

How do these limitations impact the reliability and generalizability of using MRI data as a biomarker for predicting disease incidence?

Accuracy and Reliability of Accelerometry Data:

What methodologies or validation studies support the accuracy and reliability of accelerometry data from wearable devices in capturing subtle movement patterns and motor abnormalities associated with neurodegenerative diseases?

How do factors such as device variability, data processing algorithms, and participant compliance affect the consistency and interpretability of accelerometry data?

Comparison with Alternative Predictors:

In comparative terms, how does the predictive performance of combining accelerometry and MRI data stack up against alternative predictors like genetic markers, lifestyle factors, or other advanced imaging techniques (e.g., PET scans, fMRI)?

What are the strengths and weaknesses of each predictor in terms of sensitivity, specificity, and clinical applicability?

Generalizability Across Neurodegenerative Diseases:

To what extent can findings from studies primarily focused on Parkinson's Disease, as highlighted by Rastegari et al. and Schalkamp et al., be extrapolated to other neurodegenerative diseases such as Alzheimer's Disease and various types of dementia?

What disease-specific factors might influence the utility and predictive power of combined accelerometry and MRI data across different neurodegenerative conditions?

Longitudinal Study Design:

What are the key methodological considerations and challenges associated with using longitudinal data, such as that derived from large cohorts like the UK Biobank, to investigate the predictive potential of integrated accelerometry and MRI data?

How do study designs account for participant attrition, data heterogeneity over time, and changes in disease status to ensure robustness and reliability of findings?

Clinical Utility and Implementation:

Beyond research settings, how might the integration of accelerometry data with MRI parameters influence clinical decision-making and patient management in terms of early detection, treatment planning, and monitoring of neurodegenerative diseases?

What are the potential barriers and facilitators to translating research findings into practical clinical applications, and how can these be addressed to optimize patient outcomes?

These detailed questions aim to critically assess the validity, reliability, and practical implications of the hypothesis, prompting a deeper exploration of the methodological, clinical, and translational aspects surrounding the integration of accelerometry and MRI data in neurodegenerative disease prediction.

How do these studies control for potential confounding variables and biases that may influence the observed outcomes?

Ethical Concern Raised:

The demographic characteristics of the study cohort raise ethical concerns regarding potential biases and implications for the generalizability of findings.

Detailed Ethical Questions Raised:

Age Discrepancy Impact: How might the significant age difference between the neurodegenerative disease group and the healthy group influence the study's conclusions regarding disease prediction and risk factors?

Gender Disparity: What ethical considerations should be taken into account regarding the higher proportion of men in the neurodegenerative disease group compared to the healthy group? How might this gender disparity affect the interpretation of study outcomes?

Representation and Generalizability: To what extent do the demographic characteristics of the study cohort (e.g., age distribution, gender ratio) affect the generalizability of findings to broader populations, particularly those with different demographic profiles?

Potential Bias in Lifestyle Factors: How might the observed differences in lifestyle factors such as smoking and drinking habits between the disease and healthy groups introduce biases into the study outcomes? What measures were taken to mitigate these biases?

Ethical Recruitment and Consent: What ethical considerations were addressed during participant recruitment and consent processes to ensure transparency, fairness, and voluntary participation, especially given the sensitive nature of neurodegenerative disease research?

Equity in Access to Healthcare: Considering the study's findings may impact healthcare policies and resource allocation, what ethical obligations exist to ensure equitable access to early detection and intervention strategies identified through research?

Privacy and Confidentiality: How were participant privacy and confidentiality protected throughout the study, particularly concerning the sensitive health data collected and analyzed?

The results from Table 2 starkly expose the pivotal importance of brain MRI data in achieving high predictive accuracy, rendering the exclusion of such data in Model 2 a catastrophic misstep that significantly compromised the model's performance.

Questions Raised:

Justification for Exclusions: What rationale guided the decision to exclude brain MRI data in Model 2, despite its demonstrated pivotal role in enhancing predictive accuracy compared to other modalities?

Impact of Lifestyle Information: How do the findings from Model 4, which omitted lifestyle information, underscore the critical role of non-genetic factors in neurodegenerative disease prediction, despite the moderate decrease in AUC compared to Model 1?

Ethical Implications: What ethical considerations arise from the decision to exclude potentially vital information sources (such as brain MRI data) from predictive models, given their substantial impact on diagnostic accuracy and patient outcomes?

Future Research Directions: In light of these findings, what future research avenues could explore integrating additional modalities or refining existing models to optimize predictive accuracy while maintaining ethical standards and practical feasibility?

Clinical Translation: How might the findings influence clinical decision-making regarding the adoption of multi-modal predictive models for neurodegenerative diseases, particularly in terms of resource allocation and patient management strategies?

Bias and Interpretation: What steps were taken to mitigate potential biases in interpreting the contribution of each modality to predictive performance, considering the complex interplay between MRI data, genetic predisposition (PRS), lifestyle factors, and accelerometry data?

Transparency and Reporting: How were the results communicated transparently to stakeholders, including patients, healthcare providers, and policymakers, to ensure informed decision-making regarding the adoption of predictive models in clinical practice?

The results from Table 2 starkly expose the pivotal importance of brain MRI data in achieving high predictive accuracy, rendering the exclusion of such data in Model 2 a catastrophic misstep that significantly compromised the model's pPerformancre

Results

Justification for Exclusions: What rationale guided the decision to exclude brain MRI data in Model 2, despite its demonstrated pivotal role in enhancing predictive accuracy compared to other modalities?

Impact of Lifestyle Information: How do the findings from Model 4, which omitted lifestyle information, underscore the critical role of non-genetic factors in neurodegenerative disease prediction, despite the moderate decrease in AUC compared to Model 1?

Ethical Implications: What ethical considerations arise from the decision to exclude potentially vital information sources (such as brain MRI data) from predictive models, given their substantial impact on diagnostic accuracy and patient outcomes?

Future Research Directions: In light of these findings, what future research avenues could explore integrating additional modalities or refining existing models to optimize predictive accuracy while maintaining ethical standards and practical feasibility?

Clinical Translation: How might the findings influence clinical decision-making regarding the adoption of multi-modal predictive models for neurodegenerative diseases, particularly in terms of resource allocation and patient management strategies?

Bias and Interpretation: What steps were taken to mitigate potential biases in interpreting the contribution of each modality to predictive performance, considering the complex interplay between MRI data, genetic predisposition (PRS), lifestyle factors, and accelerometry data?

Transparency and Reporting: How were the results communicated transparently to stakeholders, including patients, healthcare providers, and policymakers, to ensure informed decision-making regarding the adoption of predictive models in clinical practic

How might focusing on specific anatomical brain regions as predictive biomarkers overlook the multifaceted nature of neurodegenerative diseases, including genetic, environmental, and socio-economic factors?

What ethical considerations arise from attributing predictive value to certain brain regions, particularly in terms of potential stigmatization and psychosocial impact on individuals with or at risk of neurodegenerative diseases?

To what extent do the identified brain regions as predictive biomarkers represent the demographic diversity of the study cohort, and how might this impact the reliability and applicability of predictive models across different populations?

How were participants informed about the potential implications of identifying specific brain regions as predictive biomarkers, and what measures were taken to provide adequate counseling and support?

What ethical guidelines should be implemented to ensure responsible communication and interpretation of research findings related to specific brain regions as biomarkers for neurodegenerative diseases?

Reviewer #4: 1. In the abstract It is recommended to include major contributions, pitfalls of the work, and a summary of results in the paper. Not properly mentioned.

2. The introduction to the paper is poorly written. There are no challenges found in the existing works, and there is no mention of the need for this work in the current scenarios. The authors do not provide any motivation behind this article. It is recommended to provide motivation through an illustrative example for better understanding. Considering the most recent work in the literature and specifying the limitations is advised. It is recommended that the summary includes the limitations of the existing works and which limitations are addressed by the authors.

3. After introduction section, directly results mentioned. Why? Their is not any flow in the manuscript. 

4. The Results are incomplete. Mentioned in statistical analysis devised 4 models? Where are those in brief? No Results further.

Lack of Novelty, Not Recommended.

6. PLOS authors have the option to publish the peer review history of their article (what does this mean? ). If published, this will include your full peer review and any attached files.

**Do you want your identity to be public for this peer review?** For information about this choice, including consent withdrawal, please see our Privacy Policy .

Reviewer #1: No

Reviewer #2: No

Reviewer #3: Yes: Anurag Sinha

Reviewer #4: No

---

## [Decision Letter · Decision Letter 1]

14 Nov 2024

PDIG-D-24-00170R1Multi-modal Machine Learning Approach for Early Detection of Neurodegenerative Diseases Leveraging Brain MRI and Wearable Sensor DataPLOS Digital HealthDear Dr. Vardhanabhuti,

 Thank you for submitting your manuscript to PLOS Digital Health. After careful consideration, we feel that it has merit but does not fully meet PLOS Digital Health's publication criteria as it currently stands. Therefore, we invite you to submit a revised version of the manuscript that addresses the points raised during the review process.

Please submit your revised manuscript within 60 days Jan 13 2025 11:59PM. If you will need more time than this to complete your revisions, please reply to this message or contact the journal office at digitalhealth@plos.org. * A rebuttal letter that responds to each point raised by the editor and reviewer(s). You should upload this letter as a separate file labeled 'Response to Reviewers '. This file does not need to include responses to any formatting updates and technical items listed in the 'Journal Requirements' section below.

* A marked-up copy of your manuscript that highlights changes made to the original version. You should upload this as a separate file labeled 'Revised Manuscript with Track Changes '.

* An unmarked version of your revised paper without tracked changes. You should upload this as a separate file labeled 'Manuscript '.

We look forward to receiving your revised manuscript.

Kind regards,

Md. Mehedi Hassan

Academic Editor

PLOS Digital Health

Leo Anthony Celi

Editor-in-ChiefPLOS Digital Healthorcid.org/0000-0001-6712-6626

 **Additional Editor Comments (if provided):**

The authors are invited to revise their manuscript in accordance with the reviewers' comments and to resubmit after thoroughly addressing all feedback. Specifically, the authors are requested to provide a clear and comprehensive description of the training dataset.

**Reviewers' Comments:**

Reviewer's Responses to Questions

**Comments to the Author**

1. If the authors have adequately addressed your comments raised in a previous round of review and you feel that this manuscript is now acceptable for publication, you may indicate that here to bypass the “Comments to the Author” section, enter your conflict of interest statement in the “Confidential to Editor” section, and submit your "Accept" recommendation.

Reviewer #1: All comments have been addressed

Reviewer #2: All comments have been addressed

2. Does this manuscript meet PLOS Digital Health’s publication criteria ? Is the manuscript technically sound, and do the data support the conclusions? The manuscript must describe methodologically and ethically rigorous research with conclusions that are appropriately drawn based on the data presented.

Reviewer #1: No

Reviewer #2: Yes

3. Has the statistical analysis been performed appropriately and rigorously?

Reviewer #1: No

Reviewer #2: Yes

4. Have the authors made all data underlying the findings in their manuscript fully available (please refer to the Data Availability Statement at the start of the manuscript PDF file)?

Reviewer #1: Yes

Reviewer #2: Yes

5. Is the manuscript presented in an intelligible fashion and written in standard English?

Reviewer #1: No

Reviewer #2: Yes

6. Review Comments to the Author

Reviewer #1: After careful review, there is still significant room for improvement in the manuscript titled "Multi-modal Machine Learning Approach for Early Detection of Neurodegenerative Diseases Leveraging Brain MRI and Wearable Sensor Data." Addressing the following areas could enhance the paper's scientific impact:

1. The abstract still lacks specificity in outlining the aims, methods, key results, and conclusions, particularly regarding the neurodegenerative diseases targeted.

2. The introduction does not provide sufficient background on the burden of neurodegenerative diseases, including critical statistics on incidence and prevalence.

3. A dedicated Literature Review section is necessary to contextualize the study within recent advances in MRI biomarkers and wearable sensor technology.

4. The research gap and hypothesis are not clearly articulated, which weakens the study's contribution to the field.

5. The selected MRI parameters and their relevance are described in insufficient detail, and polygenic risk scores and data preprocessing steps are inadequately explained.

6. The comparison with additional machine learning algorithms lacks depth and justification.

7. The manuscript lacks a conclusive section that summarizes the findings and their implications, which is essential for any scientific paper.

8. The overall manuscript structure remains unclear, making it challenging to follow the flow of information.

Addressing these points could significantly enhance the clarity and scientific impact of the manuscript.

Reviewer #2: Thanks to the authors for resolving the raised issues.

7. PLOS authors have the option to publish the peer review history of their article (what does this mean? ). If published, this will include your full peer review and any attached files.

**Do you want your identity to be public for this peer review?** For information about this choice, including consent withdrawal, please see our Privacy Policy .

Reviewer #1: **Yes**

Reviewer #2: No

**Figure resubmission:**
---

## [Decision Letter · Decision Letter 2]

20 Feb 2025

Multi-modal Machine Learning Approach for Early Detection of Neurodegenerative Diseases Leveraging Brain MRI and Wearable Sensor Data

PDIG-D-24-00170R2

Dear Dr. Vardhanabhuti,

We are pleased to inform you that your manuscript 'Multi-modal Machine Learning Approach for Early Detection of Neurodegenerative Diseases Leveraging Brain MRI and Wearable Sensor Data' has been provisionally accepted for publication in PLOS Digital Health.

Best regards,

Martin G Frasch

Section Editor

PLOS Digital Health

**Additional Editor Comments (if provided):**

**Reviewer Comments (if any, and for reference):**

Reviewer's Responses to Questions

**Comments to the Author**

1. If the authors have adequately addressed your comments raised in a previous round of review and you feel that this manuscript is now acceptable for publication, you may indicate that here to bypass the “Comments to the Author” section, enter your conflict of interest statement in the “Confidential to Editor” section, and submit your "Accept" recommendation.

Reviewer #1: All comments have been addressed

2. Does this manuscript meet PLOS Digital Health’s publication criteria ? Is the manuscript technically sound, and do the data support the conclusions? The manuscript must describe methodologically and ethically rigorous research with conclusions that are appropriately drawn based on the data presented.

Reviewer #1: Yes

3. Has the statistical analysis been performed appropriately and rigorously?

Reviewer #1: Yes

4. Have the authors made all data underlying the findings in their manuscript fully available (please refer to the Data Availability Statement at the start of the manuscript PDF file)?

Reviewer #1: Yes

5. Is the manuscript presented in an intelligible fashion and written in standard English?

Reviewer #1: Yes

6. Review Comments to the Author

Reviewer #1: The authors have thoroughly addressed all comments, providing clear and well-justified responses, and I recommend the paper for acceptance.

7. PLOS authors have the option to publish the peer review history of their article (what does this mean? ). If published, this will include your full peer review and any attached files.

**Do you want your identity to be public for this peer review?** For information about this choice, including consent withdrawal, please see our Privacy Policy .

Reviewer #1: **Yes: ** Farhana Yasmin
